# Phosphorylated vimentin-triggered fibronectin matrix disaggregation enhances the dissemination of *Treponema pallidum subsp. pallidum* across the microvascular endothelial barrier

**Xi Luo**[1☯], **Litian Zhang**[1,2☯], **Xiaoyuan Xie**[1,2☯], **Liyan Yuan**[1], **Yanqiang Shi**[1], **Yinbo Jiang**[1]*, **Wujian Ke**[1]*, **Bin Yang**[1,2]*

1 Dermatology Hospital, Southern Medical University, Guangzhou, People's Republic of China, 2 The First School of Clinical Medicine, Southern Medical University, Guangzhou, People's Republic of China

☯ These authors contributed equally to this work.
* yinbo_j@126.com (YJ); kewujianstauch@163.com (WK); yangbin1@smu.edu.cn (BY)

**Data Availability Statement:** The data that support the findings of this study are available through the public repository (link: https://figshare.com/s/

## Abstract

Fibronectin (FN) is an essential component of the extracellular matrix (ECM) that protects the integrity of the microvascular endothelial barrier (MEB). However, *Treponema pallidum subsp. pallidum* (*Tp*) breaches this barrier through elusive mechanisms and rapidly disseminates throughout the host. We aimed to understand the impact of *Tp* on the surrounding FN matrix of MEB and the underlying mechanisms of this effect. In this study, immunofluorescence assays (IF) were conducted to assess the integrity of the FN matrix surrounding human microvascular endothelial cell-1 (HMEC-1) with/without *Tp* co-culture, revealing that only live *Tp* exhibited the capability to mediate FN matrix disaggregation in HMEC-1. Western blotting and IF were employed to determine the protein levels associated with the FN matrix during *Tp* infection, which showed the unaltered protein levels of total FN and its receptor integrin α5β1, along with reduced insoluble FN and increased soluble FN. Simultaneously, the integrin α5β1-binding protein–intracellular vimentin maintained a stable total protein level while exhibiting an increase in the soluble form, specifically mediated by the phosphorylation of its 39[th] residue (pSer39-vimentin). Besides, this process of vimentin phosphorylation, which could be hindered by a serine-to-alanine mutation or inhibition of phosphorylated-AKT1 (pAKT1), promoted intracellular vimentin rearrangement and FN matrix disaggregation. Moreover, within the introduction of additional cellular FN rather than other *Tp*-adhered ECM protein, *in vitro* endothelial barrier traversal experiment and *in vivo* syphilitic infectivity test demonstrated that viable *Tp* was effectively prevented from penetrating the *in vitro* MEB or disseminating in *Tp*-challenged rabbits. This investigation revealed the active pAKT1/pSer39-vimentin signal triggered by live *Tp* to expedite the disaggregation of the FN matrix and highlighted the importance of FN matrix stability in syphilis, thereby providing a novel perspective on ECM disruption mechanisms that facilitate *Tp* dissemination across the MEB.

c9eb0f390cc1063e2abb; DOI: 10.6084/m9.
figshare.26107060).

**Funding:** BY received an award by the National
Nature Science Foundation of China (URL: https://
www.nsfc.gov.cn/; Grant number: 82220108006).
WK received an award by the National Nature
Science Foundation of China (URL: https://www.
nsfc.gov.cn/; Grant number: 82072321). The
funders had no role in study design, data collection
and analysis, decision to publish, or preparation of
the manuscript.

**Competing interests:** The authors have declared
that no competing interests exist.

## Author summary

*Treponema pallidum subsp. pallidum* (*Tp*), dominating the global concerned disease—
syphilis, can mediate various pathogenic processes through its dissemination, during
which the vascular endothelial barrier plays a vital role in the pathogen-host interaction.
*Tp* alters the characteristics of the barrier components, including vascular endothelial cell
and the surrounding extracelluar matrix (ECM), while the initial step of *Tp* dissemination
involves breaching the ECM and moving toward the surface within unknown mecha-
nisms. Here, we demostrated that live *Tp* triggered the pAKT1/pSer39-vimentin signals in
microvascular endothelial cells, which promoted the redistribution of vimentin around
the perinucleus. And pSer39-vimentin facilitated the detachment of intergrein α5β1 from
fibronectin (FN), which is an essential component of ECM for *Tp* adhesion. Furthermore,
we discovered that pAKT1 inhibition or addition of cellular FN could delay the FN matrix
disaggregation, ultimately preventing live *Tp* from traversing the microvascular endothe-
lial barrier. These results provide insights into ECM disruption employed by *Tp* and iden-
tify potential therapeutic targets against syphilis.

## Introduction

Syphilis is a sexually transmitted disease, with *Treponema pallidum subsp. pallidum* (*Tp*) infec-
tion as the causative factor [1]. As the disease progresses or without proper treatment, patients
may develop skin lesions and organ-specific damage [2–4], indicating that *Tp* can traverse var-
ious microvascular endothelial barriers (MEB) to reach different organs. Previous studies have
revealed that *Tp* can enter the bloodstream of rabbits or penetrate the *in vitro* human umbilical
vein endothelial cell barrier in a matter of hours [5,6]. Currently, two predominant pathways
are proposed for microorganisms breaching the MEB: intracellular transendothelial migration,
as seen with *Streptococcus pneumoniae* [7], and the paracellular pathway, exemplified by *Neis-
seria meningitidis* disrupting cell junctions [8]. Subsequently, *Tp* was observed within cell junc-
tions and across a monolayer endothelial barrier without compromising its structural integrity
[9], and can disrupt VE-Cadherin connections and initiate lipid raft-mediated endocytosis [6].
Nonetheless, a comprehensive understanding of *Tp* dissemination remains elusive.

At the beginning of *Tp* crossing the MEB, spirochetes must initially navigate through the
extracellular matrix (ECM) to anchor themselves to the endothelial cell surface. ECM is a com-
plex network of proteins and polysaccharides, including fibronectin (FN), collagens, laminin,
and hyaluronic acid, structurally and functionally supporting the local environment surround-
ing cells. Together with the dense microvascular endothelial cells, the ECM forms an MEB
against pathogen dissemination. Recently, few studies have investigated the ECM disruption
by *Tp*, such as the fact that *Tp* can secret hyaluronidase and promoted the release of matrix
metalloproteinases [10–12], while *in vivo* experiments that can confirm these findings are lack-
ing. Cellular fibronectin (cFN) is an important glycoprotein in ECM typically secreted by cells
(such as vascular endothelial cells and fibroblasts) that forms an insoluble polymer matrix
within the MEB to maintain its stability and integrity [13], whereas the soluble plasma FN
(pFN) is primarily secreted by the liver into the bloodstream, where it circulates throughout
the body rather than forms a matrix [14]. There are distinct compositional differences between
cFN and pFN; cFN includes additional domains (EDA and EDB) compared to pFN, which can
self-assemble into fibrillar matrix together with other ECM proteins, forming insoluble FN
and insoluble matrix. Upon matrix disaggregation, cFN can become soluble FN. In contrast,

pFN circulating in the blood does not form matrix and remains soluble. These differences are crucial, as cFN provides a structural barrier that *Tp* must overcome, whereas pFN could interact with *Tp* in the bloodstream without forming such barriers. Notably, *Tp* exhibits the adhesion ability to various vital ECM components [15–17], suggesting a specific interaction that facilitates its pathogenic process. It is well-documented that *Tp* binds to fibronectin, aiding its colonization and dissemination. Previous studies identified Tp0155 and Tp0483 as FN adhesins contributing to further invasion [18], and an antibody targeting the FN-binding protein Tp0751 has shown potential in reducing *Tp* dissemination in rabbits [19]. However, the detailed mechanisms by which *Tp* interacts with FN and how this interaction influences ECM disruption and MEB penetration are still not fully understood.

As a common cytoskeletal protein that belongs to the intermediate filament family, vimentin plays a fundamental role in maintaining cell integrity and pathogeneis [20,21]. For example, *Escherichia coli K1* can induce the disassembly of vimentin polymers distributed in the cytoplasm into monomers that translocate to the cell membrane, where they interact with membrane-associated vimentin to mediate adhesion and invasive behaviors [22]. Specifically, viementin regulates the configuration of integrin α5β1 on the cellular membrane through the phosphorylation of its 39[th] residue of serine (Ser39), which in turn stabilizes the cFN matrix [23]. Because that phosphorylation at Ser39 has been implicated in altering vimentin function and its interaction with other extracellular proteins, we also focused on its potential upstream kinases. The serine/threonine kinase AKT (also known as protein kinase B subgrouped in the AGC kinase family) is a crucial regulator of various cellular processes, including metabolism, cell survival, angiogenesis, and response to various stimuli. AKT1 has been shown to phosphorylate vimentin at Ser39, thereby influencing the intergirn-FN dynamics [23,24].

Therefore, in this study, we proposed that vimentin may play an vital role in disrupting cFN-integrin binding after *Tp* infection, facilitating the dissemination of *Tp* across the MEB. Eventually, we confirmed that phosphorylated Ser39 of vimentin triggered fibronectin matrix disaggregation and the related mechanisms and domestrated that the stability of the FN matrix alleviated the dissemination of *Tp* across the MEB.

## Methods and materials

### Ethics statement

All procedures mentioned above were approved by the Medical Ethics Committee of Dermatology Hospital of Southern Medical University and the Animal Ethics Committee of South China Agricultural University (2021c036).

### Cell culture

Wild-type and stable-transfected HMEC-1 (human dermal microvascular endothelial cell-1) were cultivated in completed endothelial cell medium (1001; ScienCell, USA) in a humidified atmosphere at 37˚C and 5% $CO_2$, while HEK-293T cells for lentivirus packaging in DMEM medium (C11995500BT; Gibco, USA) with 10% (v/v) fetal bovine serum (FBS) (F8318; Sigma-Aldrich, USA).

### Lentivirus packaging

HEK-293T cells in the logarithmic growth phase were seeded in 10 cm culture dishes at a density of 70%. When the cell density approached 90%, cells were subjected to serum starvation by culturing for 2 hours in DMEM medium without FBS. Liposome mixtures were prepared according to the manufacturer's instructions of Lipofectamine 3000 Transfection Kit

(L3000015; Invitrogen, USA): Tube A contained 500.0 µL Opti-MEM medium (31985070; Gibco, USA) and 20.0 µL Lipo3000; and Tube B contained 500.0 µL Opti-MEM medium, 4.0 µg pLV3-CMV-vimentin (human)-3×flag-puro plasmid (OE-S39-Vim) or 4.0 µg pLV3-CMV-vimentin (human)-S39A-3×flag-puro plasmid (OE-S39A-Vim), 3.0 µg psPAX2 plasmid, 1.0 µg pDM2.G plasmid, and 16.0 µL P3000. Subsequently, cells were replaced with 6.0 mL of fresh DMEM medium with 10% (v/v) FBS, and the liposome mixture was added dropwise to the medium. After 6 hours, the entire culture medium was discarded and replaced with 8.0 mL of DMEM medium with 10% (v/v) FBS for cultivation.

After 48 hours post-transfection, the entire medium was collected and filtered through 0.45 µm syringe driven-filters (FPE404030; JET BIOFIL, China). The filtrate was mixed with 5× virus precipitation buffer (NaCl 8.766 g; PEG8000 50.0 g; ultrapure water 200.0 mL) at a ratio of 4:1 and incubated overnight at 4˚C. Then, the mixture was centrifuged at 4,000 ×g for 20 minutes at 4˚C. The supernatant was discarded, and the lentivirus precipitate was dissolved by 200.0 µL of pre-cooled PBS, which was divided into aliquots of 50.0 µL per tube and stored at -80˚C for future use.

## Establishment of stable-transfected HMEC-1 cell lines

HMEC-1 cells in the logarithmic growth phase were seeded in 6-well culture plates at a density of 50%. A 50.0 µL lentivirus suspension (OE-S39-Vim or OE-S39A-Vim) and 1.0 µL polybrene (40804ES76; Yeasen, China) were added into 2.0 mL culture medium per well, followed by the incubation for 24 hours. Subsequently, the transfected cells were cultured with the medium containing 1.0 µg/mL puromycin (60209ES10; Yeasen, China) for 5-day selection, while with the maintenance medium containing 0.25 µg/mL puromycin to ensure the overexpression efficiency of the stable-transfected HMEC-1 cell lines.

## Preparation and enumeration of *Tp* Nichols strain

The *Tp* Nichols strain was provided by Prof. Tiebing Zeng from University of South China. This strain was propagated with the rabbit infectivity test and resuspended to obtain live *Tp* suspension referred the previous studies [25,26]. To obtain an inactivated *Tp* suspension, 1% (v/v) penicillin-streptomycin was added for a 30-minute incubation. The number of *Tp* organisms was determined using the dark field microscopy (DFM) enumeration method [25,27]:

A ten-microliter (10.0 µL) of *Tp* suspension was placed on a slide and covered with a $22.0 \times 22.0$ mm coverslip, providing an area of 4.84 cm$^2$ ($22.0$ mm $\times 22.0$ mm $= 484$ mm$^2 = 4.84$ cm$^2$) and a thickness of 0.0207 mm (10.0 µL $= 0.01$ cm$^3$, 0.01 cm$^3$/4.84 cm$^2 = 0.00207$ cm $= 0.0207$ mm).

Spiral-shaped *Tp* organisms were counted in twenty to fifty of microscopic fields (40×) when each field contained more than 10 organisms; otherwise, at least one hundred fields were counted (recorded as $X_1$, $X_2$, . . ., $X_n$, where n represented the number of fields).

In each field, the radius (r) of the DFM (Olympus, JAPAN; BX43) under the 40× microscope is 0.205 mm, resulting in a field area of 0.134 mm$^2$ ($\pi r^2$), and a volume of $2.73 \times 10^{-6}$ cm$^3$ (0.134 mm$^2 \times 0.0207$ mm $= 2.73 \times 10^{-3}$ mm$^3 = 2.73 \times 10^{-6}$ cm$^3$).

Therefore, the concentration (organisms/mL) was calculated using the formula: $(X_1 + X_2 + . . . + X_n)/(2.73 \times 10^{-6} \times n)$.

## Detection of viability and motility of *Tp*

HMEC-1 cells ($1 \times 10^6$/mL, 200.0 µL per well) were added to a 24-well plate (containing a cell slide in each well) and incubated for 24 hours. Following this, live *Tp* was co-cultured with the

cells for 6 and 8 hours. After the established time, the cell slides were removed and examined under the DFM to observe the viability and motility of *Tp*. Videos and photographs were recorded to document the observations.

### *In vitro* infection with *Tp*

HMEC-1 cells were seeded in 6-well plates ($1 \times 10^6$ cells per well/per coverslip) overnight and treated with TpCM-2 [28] (the control group), live *Tp* suspension (the Ltp group), or inactivated *Tp* suspension (the Dtp group), as well as pretreated with Capivasertib (1.0 μM for 1 hour) (pAKT1 inhibitor; HY-15431; MedChem Express, USA), cFN protein (2.5 μg for 1 hour) (F0556; Sigma-Aldrich, USA), pFN (2.5 μg for 1 hour) (F2006; Sigma-Aldrich, USA), or laminin (2.5 μg for 1 hour) (F4544; Sigma-Aldrich, USA).

### Detection of adhesion ability of *Tp* to ECM proteins

cFN, pFN, and laminin (1.25 μg of each protein per 100.0 μL PBS per well) were added to a 96-well plate and incubated at 37°C for 2 hours. After incubation, the supernatant was discarded, and the plate was air-dried in a safety cabinet. Subsequently, 50.0 μL of live *Tp* suspension (containing $1 \times 10^4$ organisms) was added to each well and incubated at 37°C for 6 hours. The supernatant from each well was then collected, and the wells were washed five times with 100.0 μL PBS. All the PBS and supernatant from each well were combined and centrifuged and centrifuged at 16,000 ×g for 8 minutes. The precipitates containing nonadherent *Tp* were resuspended with PBS and enumerated using DFM.

### Immunofluorescence assay (IF)

HMEC-1 cells ($1 \times 10^6$/mL, 200.0 μL per well) plated on the Chamber Slides (177380; Thermo Fisher, USA) were fixed with 4% paraformaldehyde (P0099; Beyotime, China) for 15 minutes at room temperature (RT), permeabilized with 0.5% (v/v) Triton X-100 (ST1722; Beyotime, China) for 15 minutes at RT, and blocked in 2% (w/v) bovine serum albumin (BSA; 36101ES76l; Yeasen, China) for 1 hour. Then, cells were incubated with primary antibodies overnight at 4°C and subsequently with fluorescence-conjugated secondary antibodies for 1 hour at RT. The nuclei were counterstained with 300.0 nM DAPI (C1002; Beyotime, China) for 2 minutes. Fluorescent-stained images were taken using a confocal microscope (Nikon, Japan).

Primary antibodies included: Anti-Fibronectin antibody (1:50; ab281574; Abcam, USA); Anti-Treponema pallidum antibody (1:3000; ab20923; Abcam, USA); Anti-Vimentin antibody (1:100; ab92547; Abcam, USA); Anti-F-actin antibody (1:200; ab130935; Abcam, USA). Secondary antibodies included: Goat Anti-Rabbit IgG H&L (Alexa Fluor 488) (1:500; ab150077; Abcam, USA); Goat Anti-Rabbit IgG H&L (Alexa Fluor 594) (1:500; ab150080; Abcam, USA); Goat Anti-Mouse IgG H&L (Alexa Fluor 488) (1:500; ab150113; Abcam, USA); Goat Anti-Mouse IgG H&L (Alexa Fluor 594) (1:500; ab150116; Abcam, USA).

### Western Blotting (WB)

Total proteins extraction was performed referred to the previous study [29], and the concentration of total proteins was examined by the BCA Protein Assay Kit (P0011; Beyotime, China) according to the manufacturer's instructions. And the extraction of insoluble and soluble FN/vimentin was performed by the DOC solubility assay [30,31]: Cells were lysed in the DOC lysis buffer (2% DOC [S579505; Aladdin, China]; 20.0 mM Tris-HCl, pH 8.8; 2.0 mM N-ethylmaleimide; 2.0 mM iodoacetic acid; 2.0 mM EDTA; and 2.0 mM phenylmethylsulfonyl fluoride

[PMSF]), passed through a 26-gauge needle, and centrifuged at 18,400 ×g for 20 minutes at 4˚C. The DOC-soluble fraction (supernatant) was retained and determined for protein concentrations by the BCA Protein Assay Kit, while the DOC-insoluble pellet was dissolved in SDS-solubilization buffer (1% [w/v] SDS; 20.0 mM Tris-Cl, pH 8.8; 2.0 mM EDTA; 2.0 mM iodoacetic acid; 2.0 mM N-ethylmaleimide; and 2.0 mM PMSF).

One hundred (100.0) μg protein per lane was loaded on 6% (w/v) or 10% SDS-PAGE gels and transferred to 0.45 μm PVDF membranes. Membranes were blocked with 5% (w/v) BSA for 1 hour, incubated overnight at 4˚C with primary antibodies, subsequently incubated with HRP-conjugated secondary antibodies for 1 hour at RT, and visualized using chemiluminescence (Bio-Rad, USA). The signal intensities were quantified using the Image J software [32].

Primary antibodies included: Anti-Integrin alpha5 antibody (1:5000; ab150361; Abcam, USA); Anti-Integrin beta1 antibody (1:2000; ab30394; Abcam, USA); Anti-Fibronectin antibody (1:500–1:1000; ab281574; Abcam, USA); Anti-GAPDH antibody (1:5000; ab8245; Abcam, USA); Anti-Vimentin antibody (1:1000; ab8069; Abcam, USA); Phospho-Vimentin (Ser39) Antibody (1:1000; 13614; Cell Signaling Technology, USA); Anti-Vimentin (phospho S56) antibody (1:1000; ab217673; Abcam, USA); Anti-Vimentin (phospho S72) antibody (1:5000–1:10000; ab52944; Abcam, USA); Phospho-Vimentin (Ser83) Antibody (1:1000; 12569; Cell Signaling Technology, USA); Anti-α-tubulin antibody (1:10000; ab7291; Abcam, USA); AKT1 (C73H10) Rabbit mAb (1:1000; 2938; CST, USA); Anti-AKT1 (phospho S473) antibody (1:5000–1:10000; ab81283; Abcam, USA). Secondary antibodies included: Goat Anti-Rabbit IgG H&L (HRP) (1:5000; ab6721; Abcam, USA); Goat Anti-Mouse IgG H&L (HRP) (1:5000; ab205719; Abcam, USA).

## Co-immunoprecipitation experiment (co-IP)

Cell lysates were divided into two parts: 10% of the total lysate was reserved as the input control to collect total proteins and measure the concentration, while the remaining lysate was used for immunoprecipitation (IP), which was incubated overnight at 4˚C with either an anti-Vimentin antibody (1:200; ab137321; Abcam, USA), an anti-AKT1 antibody (1:200; ab235958; Abcam, USA), or control IgG. The next day, the IP lysates were enriched with magnetic Protein A/G beads (P2179; Beyotime, China) following the manufacturer's instructions. Finally, the input and IP collections were subjected to western blotting analysis.

## Endothelial barrier traversal experiment

HMEC-1 cells ($1 \times 10^6$/mL, 200.0 μL per well) were added to the upper chamber of a 24-well Transwell plate (3422; Corning, USA) with a pore diameter of 8 μm to form a dense monolayer overnight, followed by cFN/pFN/laminin pretreatment (2.5 μg for 1 hour, respectively) and live *Tp* infection. After 24 hours, the culture medium in the lower chamber was collected for DNA quantitative analysis.

## *In vivo Tp*-challenged experiment

Three adult male New Zealand white rabbits (2.5–3.5 kg) per group were housed individually at 18–20˚C and given antibiotic-free food and water. Rabbits were inoculated intradermally at eight sites on their clipped backs with 100.0 μL of suspension containing 2.5 μg of cFN, pFN, laminin, or PBS per site, immediately followed by in situ inoculation with 100.0 μL of live *Tp* ($1 \times 10^6$ /mL). Skin biopsy and blood collection were performed every 3 days in the first week after *Tp* inoculation; and subsequently, blood collection and lesion monitoring were performed every 4 days, followed by organ (cutaneous lesion/liver/spleen/lymph node) collection

performed in the day when the cutaneous lesion in the PBS+live *Tp* group began to ulcerate or the TRUST titer in this group turned positive.

## DNA extraction from cell culture medium and rabbit tissue

DNA was extracted from cell culture medium and rabbit tissue using the DNeasy Blood & Tissue Kit (69504; Qiagen, Germany), following the manufacturers' instructions. For DNA extraction from cell culture medium, samples were added 200.0 μL AL buffer and 20.0 μL Proteinase K, followed by vortexing thoroughly and incubation at 56°C for 30 minutes. And for DNA extraction from *Tp*-challenged rabbits, samples of skin (20.0 mg per biopsy, 2 biopsies per rabbit), liver (20.0 mg per biopsy, 2 biopsies per rabbit), spleen (10.0 mg per biopsy, 2 biopsies per rabbit), and lymph node (an entire one from the popliteal fossa per rabbit) were transferred into 15 mL tube containing cool PBS and then digested with 200.0 μL AL buffer and 20.0 μL Proteinase K, followed by vortexing thoroughly and incubation at 56°C for 16 hours. DNA precipitation was washed in accordance with the manufacturer's protocol, and DNA elution was performed twice with one aliquot of 120.0 μL elution buffer.

## Quantitative analysis by real-time PCR (qPCR) for *Tp* burden

Real-time PCR was performed to detect the absolute quantitative copies of *Tp polA* and rabbit *β-actin* [26,33], for which the *Tp polA* qPCR standard and rabbit *β-actin* qPCR standard with ten-fold serial dilutions were also prepared as described. The sequences of primers and probes were as follows: *Tp polA* F' primer (5'→3') CAGGATCCGGCATATGTCC; *Tp polA* R' primer (5'→3') AAGTGTGAGCGTCTCATCATTCC; *Tp polA* probe (5'→3') 6FAM-CTGTC ATGCACCAGCTTCGACGTCTT-BHQ1; Rabbit *β-actin* F' primer (5'→3') TGGCTCTAAC AGTCCGCCTAG; Rabbit *β-actin* R' primer (5'→3') AGTGCGACGTGGACATCCG; Rabbit *β-actin* probe (5'→3') 6FAM-CGAGTCGGGCCCCTCCATCGTGCACCGCAA-BHQ1. And the total 25.0 μL of PCR reaction volume was prepared as follows, avoiding light exposure: 12.5 μL TaqMan Gene Expression Master Mix (4369016; Thermo Fisher, USA), 1.0 μL $MgCl_2$ (50 mM) (AM9530G; Thermo Fisher, USA), 2.5 μL 10× Primer Mix (2.5 μM), 2.5 μL 10× probe (2.0 M), 1.5 μL DNase/RNase-free Distilled water (10977015; Thermo Fisher, USA), and 5.0 μL DNA template. Thermocycling was performed in a Bio-Rad CFX384 system as follows: 1 cycle of 95°C for 10 minutes, followed by 50 cycles of 95°C for 15 seconds and 60°C for 1 minute. The standard curves were generated automatically by the Bio-Rad system, based on which the gene copies of each sample could be calculated. And the *Tp polA* copies should be normalized by rabbit *β-actin* (for rabbit tissues) or by the volume of culture medium (for the endothelial barrier traversal experiment).

## Serological tests

Serum samples were separated from 5.0 mL peripheral blood of each rabbit, followed by centrifugation of 3,000 rpm for 10 minutes. And the sera were monitored with Treponema pallidum particle agglutination (TPPA) test (SERODIA-TP•*PA* Kit; 1633; Fujirebio Inc., Japan) and the toluidine red unheated serum (TRUST) test (TRUST Kit; S10940058; Rongsheng Bio., China), following the manufacturers' instructions.

## Statistical analysis

Values were the means ± SDs of triplicate samples and representative of three independent experiments. One-way ANOVA and student's t test were used to compare values among multiple groups and two groups, respectively. p-value < 0.05 represents statistical difference.

## Results

### Live *Tp* disaggregated FN matrix around microvascular endothelial cells

Localized colonization and subsequent rapid dissemination of *Tp* in the skin are intricately connected with the microvasculature. Notably, when observed by fluorescent microscopy, the clustered FN matrix around HMEC-1 cells diminished after 6 hours of live *Tp* (Ltp group) stimulation compared with that in the control group (Ctrl group); however, no such alteration was observed with an equivalent amount of inactivated dead *Tp* (Dtp group)stimulation (**Fig 1A**). Subsequently, the motility and viability of *Tp* after 6-hour and even 8-hour stimulation to HMEC-1 was detected using DFM and *Tp*-infected rabbit model (**S1 Fig and** S1 **and** S2 **Movies**). The findings revealed the benign motility of *Tp* in the Ltp group during the stimulation period and the retained pathogenicity of nonadherent *Tp* in this group to develop primary syphilitic lesions after intradermal injection in rabbits, suggesting that within the short co-culture period, *Tp* retained its capacity to interact with endothelial cells in a manner likely consistent with early stages of infection.

Given the proposed pathway for ECM stabilization involving cFN dimer, transmembrane integrin, and intracellular vimentin, we hypothesized that the disappearance of the FN matrix could be associated with a decrease in the levels of FN protein and integrin α5β1 protein on HMEC-1 cells. Therefore, upon comparing the protein levels in the three groups at 0 and 6 hours, no difference was determined in the total FN and integrin α5β1 protein content (**Fig 1B**). Results obtained from co-culturing HMEC-1 cells with viable *Tp* at different multiplicities of infection (MOI) (**Fig 1C**) and cultivating for varied durations (**Fig 1D**) revealed that the levels of total FN and integrin α5β1 remained stable. Previous studies have indicated that the formation of FN matrix results from the aggregation of insoluble FN assembled by soluble individual FN molecules [30,34]. This suggests that the decrease in the FN matrix could potentially be associated with its polymerization. In the Ltp group, the levels of soluble FN were significantly higher than those in the other two groups, as determined by extracting soluble and insoluble FN separately, whereas the levels of insoluble FN were markedly lower (**Fig 1E**). Furthermore, soluble FN increased with increasing live *Tp* MOIs or prolonged stimulation time, whereas insoluble cFN decreased (**Fig 1F–1G**). This demonstrated that live *Tp* directly induced FN matrix disaggregation around microvascular endothelial cells, mediating by altering the polymerization state of FN rather than in a degradative manner.

### *Tp*-induced vimentin phosphorylation and rearrangement promoted disaggregation of the FN matrix

According to well-established evidence, the phosphorylation of serine residues in intracellular vimentin enhances vimentin's solubility, thereby hindering the translocation of vimentin from the perinucleus to the inner cell membrane [23]. HMEC-1 cells was co-cultured with live *Tp* (Ltp group) at the indicated MOIs for 6 hours; no significant difference in the protein levels of total vimentin was observed in Western blotting analysis; however, the soluble vimentin level increased as the MOIs elevated; insoluble vimentin decreased, notably at MOI ≥ 1 (**Fig 2A**). Similarly, the Ltp group exhibited a time-dependent trend, with significant changes occurring after ≥ 1 hour (**Fig 2B**). Evidently, only live *Tp* stimulation induced the above-mentioned alteration (**Fig 2C**). The results of immunofluorescence revealed that live *Tp* confined vimentin distribution around the nucleus, in contrast to the Ctrl and Dtp groups, in which vimentin was uniformly distributed throughout the cells (**Fig 2D**).

Subsequently, an analysis of phosphorylation changes occurring at four serine residues (Ser39, Ser56, Ser72, and Ser83) on vimentin revealed that only the phosphorylated

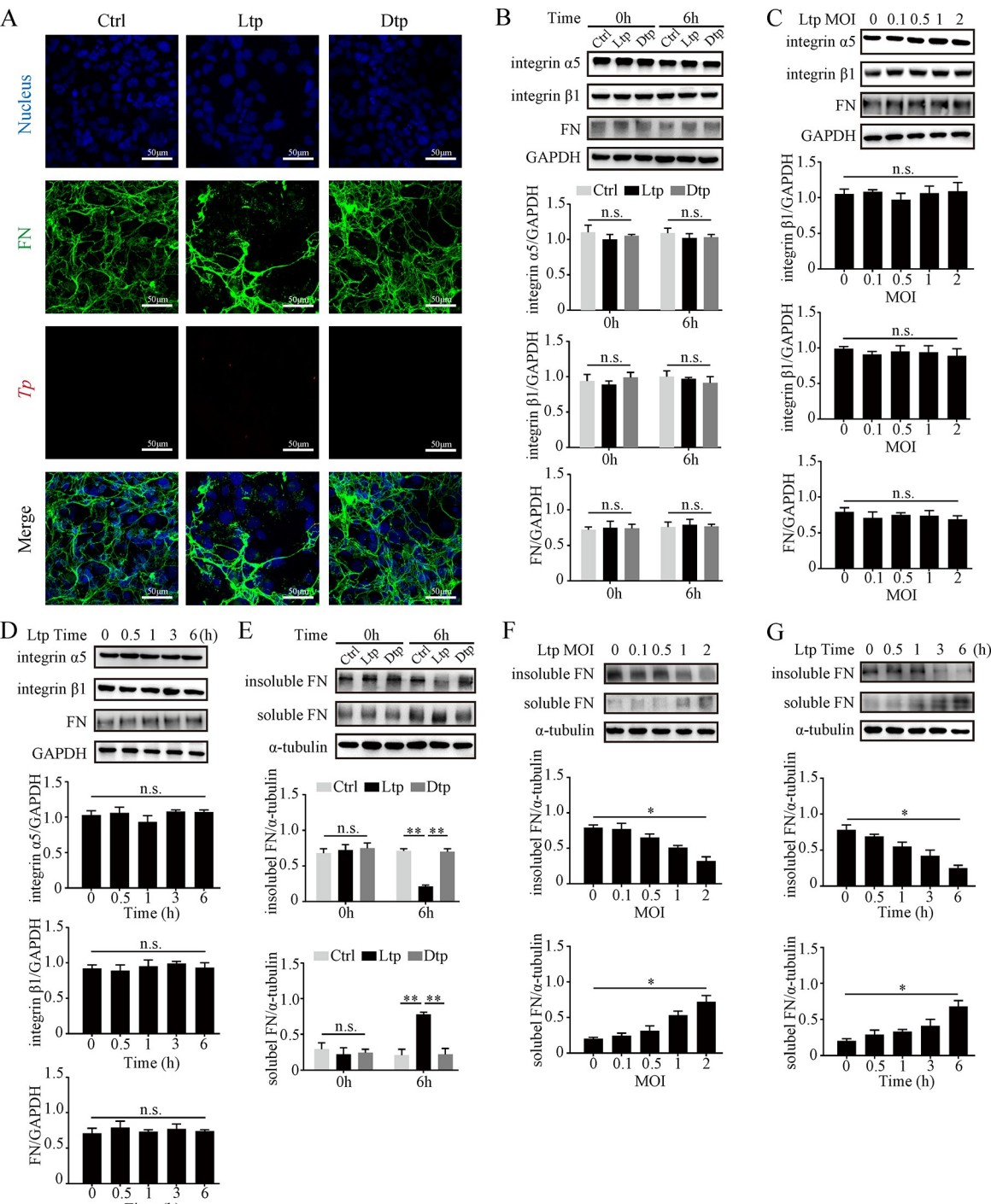

**Fig 1. Live *Tp* disaggregated FN matrix around microvascular endothelial cells. (A)** FN matrix around HMEC-1 after 6 hours of stimulation with Ctrl, live *Tp* (MOI 2), and dead *Tp* (MOI 2); as observed by fluorescence microscopy; blue for the nucleus, green for the FN matrix, and red for *Tp*, scale bar = 50 μm. **(B)** Protein expressions of integrin α5, integrin β1, and total FN in HMEC-1 after 0 and 6 hours of stimulation with Ctrl, live *Tp* (MOI 2), and dead *Tp* (MOI 2). **(C, D)** Protein expressions of integrin α5, integrin β1, and total FN in HMEC-1 (C) after 6 hours of stimulation with live *Tp* at various MOIs (0, 0.1, 0.5, 1, and 2) and (D) after stimulation with live *Tp* (MOI 2) for different durations (0, 0.5, 1, 3, and 6 hours). **(E)** Protein expressions of insoluble FN and soluble FN in HMEC-1 after 0 and 6 hours of stimulation with Ctrl, live *Tp* (MOI 2), and dead *Tp* (MOI 2). **(F, G)** Protein expressions of insoluble FN and soluble FN in HMEC-1 (F) after 6 hours of stimulation with live *Tp* at various MOIs and (G) after stimulation with live *Tp* for different durations. Ctrl: negative control; Ltp: live *Tp*; Dtp: dead *Tp*; FN: fibronectin; n.s.: no significance; *: p value < 0.05; **: p value < 0.01.

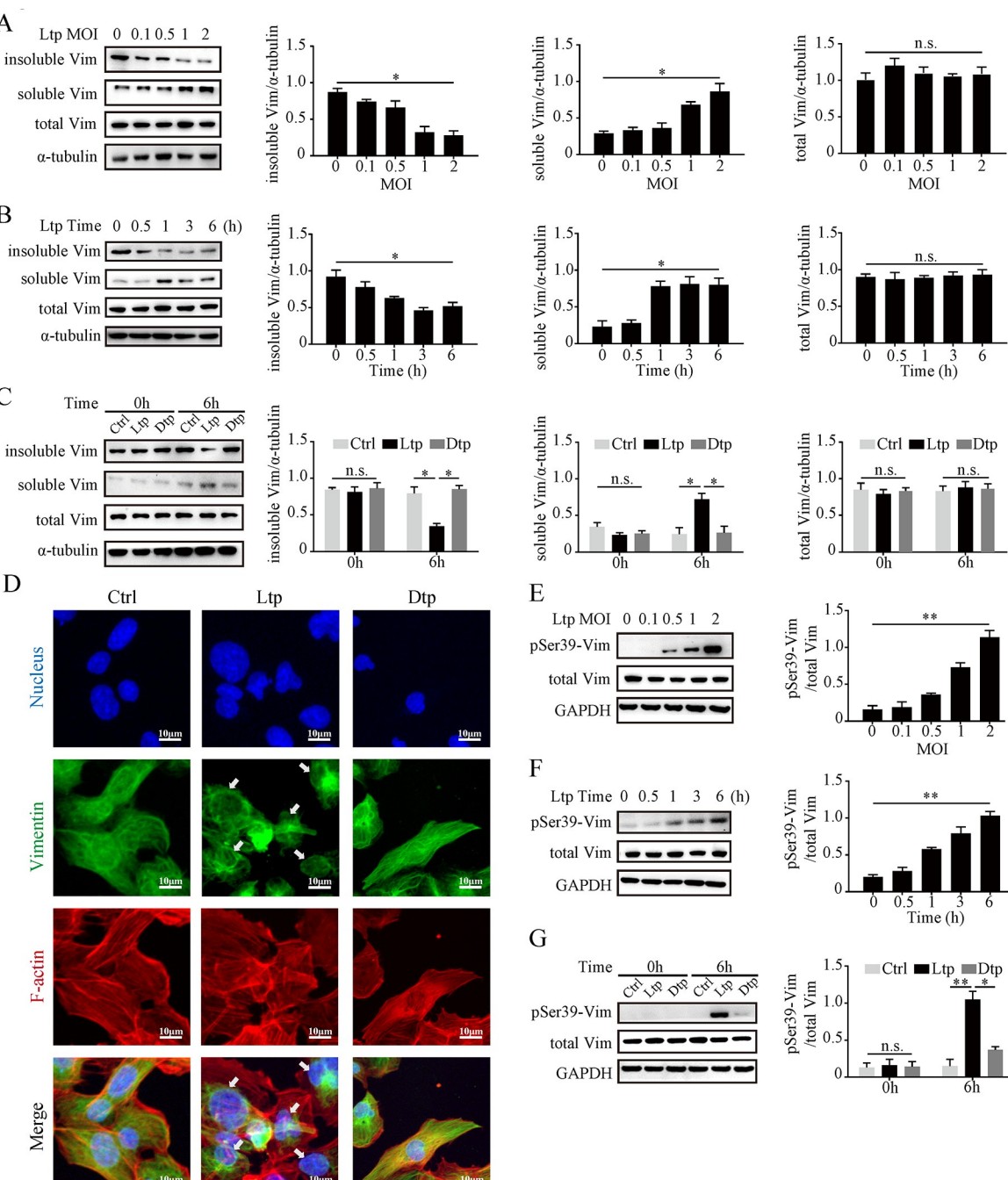

**Fig 2. *Tp* induced the conversion of insoluble and soluble vimentin protein, leading to its rearrangement and Ser39 residue phosphorylation. (A, B)** Protein expressions of insoluble vimentin, soluble vimentin, and total vimentin in HMEC-1 (A) after 6 hours of stimulation with live *Tp* at various MOIs (0, 0.1, 0.5, 1, and 2) and (B) after stimulation with live *Tp* (MOI 2) for different durations (0, 0.5, 1, 3, and 6 hours). **(C)** Protein expression of insoluble vimentin, soluble vimentin, and total vimentin in HMEC-1 after 0 and 6 hours of stimulation with Ctrl, live *Tp* (MOI 2), and dead *Tp* (MOI 2). **(D)** The vimentin arrangement in HMEC-1 after 6 hours of stimulation with Ctrl, live *Tp* (MOI 2), and dead *Tp* (MOI 2), as observed by fluorescence microscopy; blue for the nucleus, green for vimentin, red for F-actin, scale bar = 10 μm. **(E, F)** Protein expressions of pSer39-vimentin and total vimentin in HMEC-1 (E) after 6 hours of stimulation with live *Tp* at various MOIs and (F) after stimulation with live *Tp* (MOI 2) for different durations. **(G)** Protein expressions of pSer39-vimentin and total vimentin in HMEC-1 after 0 and 6 hours of stimulation with Ctrl, live *Tp*, and dead *Tp*. Ctrl: negative control; Ltp: live *Tp*; Dtp: dead *Tp*; Vim: vimentin; n.s.: no significance; *: p value < 0.05; **: p value < 0.01.

Ser39-vimentin (pSer39-vimentin) markedly augmented with ascending MOI or co-culture duration (**Fig 2E–2F**). This distinct change was exclusively observed in the Ltp group (**Fig 2G**), as no differences were detected at the remaining residues (**S2 Fig**). Additionally, a vimentin overexpression vector (OE-S39-Vim) and a mutated vector (Serine to alanine; OE-S39A--Vim) were constructed (**Fig 3A**) and stably transfected into HMEC-1 cells, respectively. Co-culturing with live *Tp* uncovered that HMEC-1$^{\text{OE-S39A-Vim}}$ boosted insoluble FN and considerably reduced soluble FN in comparison to HMEC-1$^{\text{WT}}$ and HMEC-1$^{\text{OE-S39-Vim}}$ (**Fig 3B**). The results of immunofluorescence manifested that HMEC-1$^{\text{OE-S39A-Vim}}$ did not exhibit FN matrix disaggregation (**Fig 3C**). Moreover, the endothelial barrier traversal experiment indicated that the *Tp* burden in the lower chamber of the OE-S39A-Vim group was significantly lower compared to the other two groups (**Fig 3D–3E**). These results confirmed that *Tp* induced

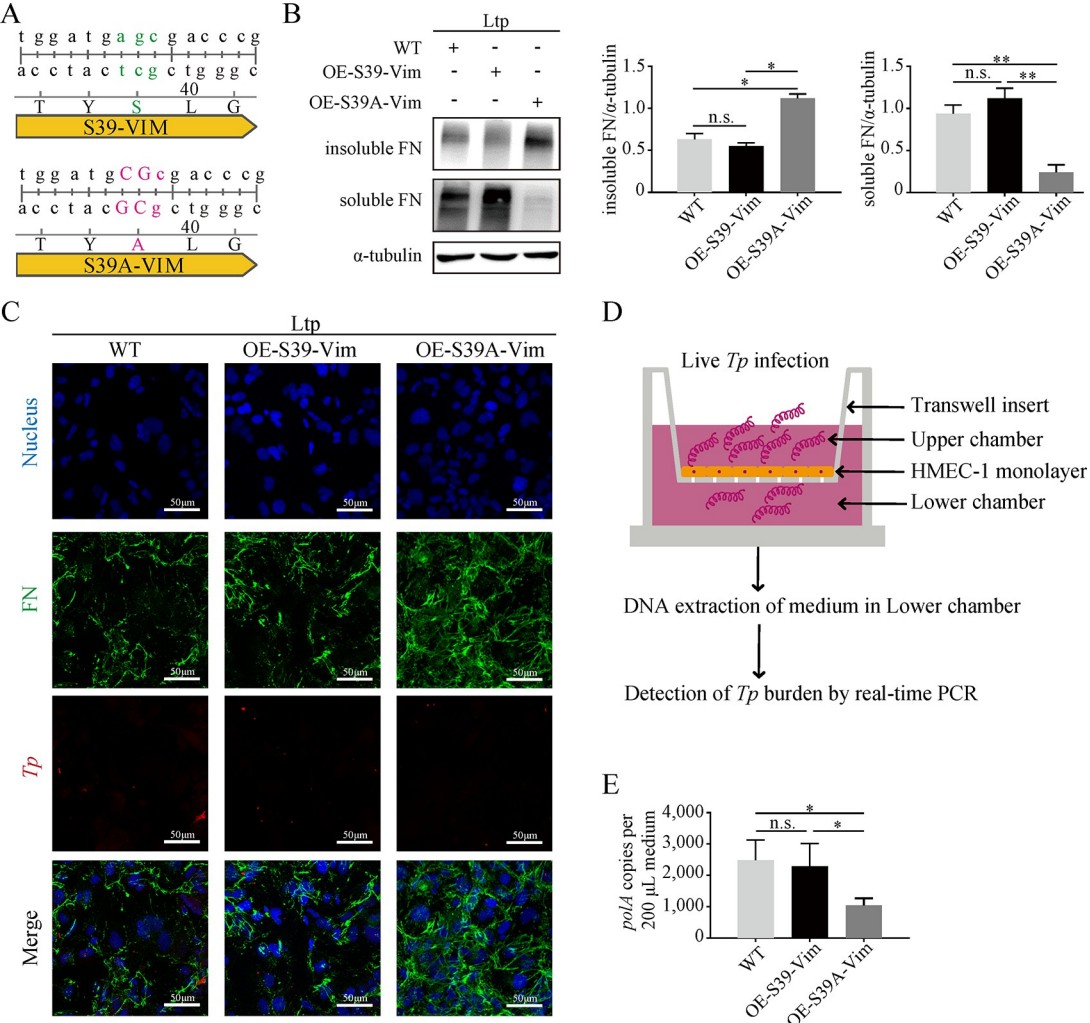

**Fig 3. *Tp* mediated the disaggregation of the FN matrix by promoting Ser39 residue phosphorylation of vimentin. (A)** Construction of overexpressed S39-Vimentin (OE-S39-Vim) and S39A-Vimentin (OE-S39A-Vim) lentiviral vectors. **(B)** Protein expressions of insoluble FN and soluble FN in HMEC-1$^{\text{WT}}$, HMEC-1$^{\text{OE-S39-Vim}}$, and HMEC-1$^{\text{OE-S39A-Vim}}$ after 6 hours of stimulation with live *Tp* (MOI 2). **(C)** FN after co-culturing *Tp* (MOI 2) and HMEC-1 for 6 hours, as observed by fluorescence microscopy; blue for the nucleus, green for the FN matrix, and red for *Tp*, scale bar = 50 μm. **(D)** Schematic diagram of the endothelial barrier traversal experiment; after the formation of HMEC-1 monolayer, live *Tp* (MOI 2) added for stimulation. **(E)** Penetrated spirochete burden (copies of *Tp polA*) of all culture medium (200.0 μL) in the lower chamber; detected by qPCR. WT: wild type; Ctrl: negative control; Ltp: live *Tp*; Dtp: dead *Tp*; Vim: vimentin; FN: fibronectin; n.s.: no significance; *: p value < 0.05; **: p value < 0.01.

pSer39-vimentin and its intracellular rearrangement in HMEC-1 cells, thereby mediating the disaggregation of the FN matrix.

## *Tp* facilitated the phosphorylation of Ser39-vimentin via the pAKT1 pathway

Obviously, follow-up explorations into how *Tp* activates vimentin phosphorylation would contribute to a comprehensive understanding of the molecular mechanisms underlying *Tp* dissemination. Furthermore, the Group-based Prediction System 6.0 (https://gps.biocuckoo.cn/) was utilized to identify potential serine/threonine phosphorylation sites on vimentin and their associated protein kinases [35]. The results revealed that the Ser39 residue was most likely phosphorylated by AGC kinases (**Fig 4A** and **S1 Table**). This finding aligned with a previous investigation suggesting that pSer39-vimentin can be mediated by AKT1 [24]. To validate this statement, the level of phosphorylated AKT1 (pAKT1) in HMEC-1 cells was examined, which unveiled an increased pAKT1 level with higher MOIs or longer durations (**Fig 4B–4C**). Compared to the other groups, the pAKT1 level in the Ltp group was significantly higher (**Fig 4D**), while total AKT1 expression remained unchanged in all these experiments. Subsequently, the physical interaction between vimentin and AKT1 was demonstrated using co-immunoprecipitation (co-IP) experiments (**S3 Fig**). Moreover, the upregulation of pAKT1 in HMEC-1 cells induced by live *Tp* stimulation was effectively reversed by pre-treatment with the AKT1 inhibitor Capivasertib for 1 hour (**Fig 4E**), which also profoundly increased the level of insoluble FN and the inhibition of soluble FN (**Fig 4F**). The Ltp+Capivasertib group did not exhibit distinct extracellular FN matrix disaggregation, as opposed to the Ltp group (**Fig 4G**). These results suggested that attenuated AKT1 kinase activity effectively prevented *Tp*-induced pSer39-vimentin and restrained the subsequent disaggregation of the FN matrix.

## Dense FN matrix attenuated *Tp* penetration through the trans-endothelial barrier *in vitro*

In the endothelial barrier traversal experiment, different types of ECM proteins (including cFN, pFN, and laminin) that had been previously reported to adhere to *Tp* [15] were utilized to investigate their impact on *Tp* penetration (**Fig 5A**). One hour before live *Tp* stimulation (MOI 2), ECM proteins were added to the HMEC-1 monolayer. The penetrating *Tp* burden in the lower chamber was the lowest in the cFN+Ltp group, whereas the pFN+Ltp and Laminin +Ltp groups showed burdens similar to the Ltp group (**Fig 5B**). The same grouping and procedure were performed on chamber slides without permeabilizing the samples, as displayed in **Fig 5C**. This ensured that immunofluorescence corresponded entirely to the extracellular states of FN and *Tp* (mostly punctate or short rod-shaped in red color) around the cell surface. In contrast to the FN pattern observed in the Ctrl group, the Ltp group exhibited noticeable gaps in the cFN matrix, with cord-like cFN aggregates mainly disappearing. Meanwhile, the cFN+Ltp group demonstrated a more tightly packed reticular FN structure with the emergence of numerous small filamentous aggregates. The Laminin+Ltp group displayed a minor difference compared with the Ctrl group, with only a slight decrease in the fluorescence intensity of FN. A similar trend was observed in the pFN+Ltp group, but the matrix structure became sparse. In particular, among the four groups with live *Tp* stimulation, the cFN+Ltp group had the largest quantity of *Tp* around the cell surface, whereas the remaining groups showed comparable amounts. On the other hand, the ahesion capability of *Tp* to cFN, pFN, and laminin was determined, which was utilized the medium after after *Tp*'s adhesion to these proteins for 6 hours to collect and enumerate the nonadherent *Tp* (**S4 Fig**). Surprisingly, our results demonstrated that *Tp* exhibits maximal adherence to laminin instead of cFN. These existing data

suggested that the structural integrity of the FN matrix plays a critical role in preventing *Tp* invasion, potentially independent of direct *Tp*-ECM adhesion interactions.

## Cellular FN inhibited the cutaneous in situ dissemination of *Tp* and its organic infection *in vivo*

Our *in vitro* experiments highlighted the importance of maintaining FN matrix integrity to impede the *Tp* dissemination. As a fairly high homology between human and rabbit FN

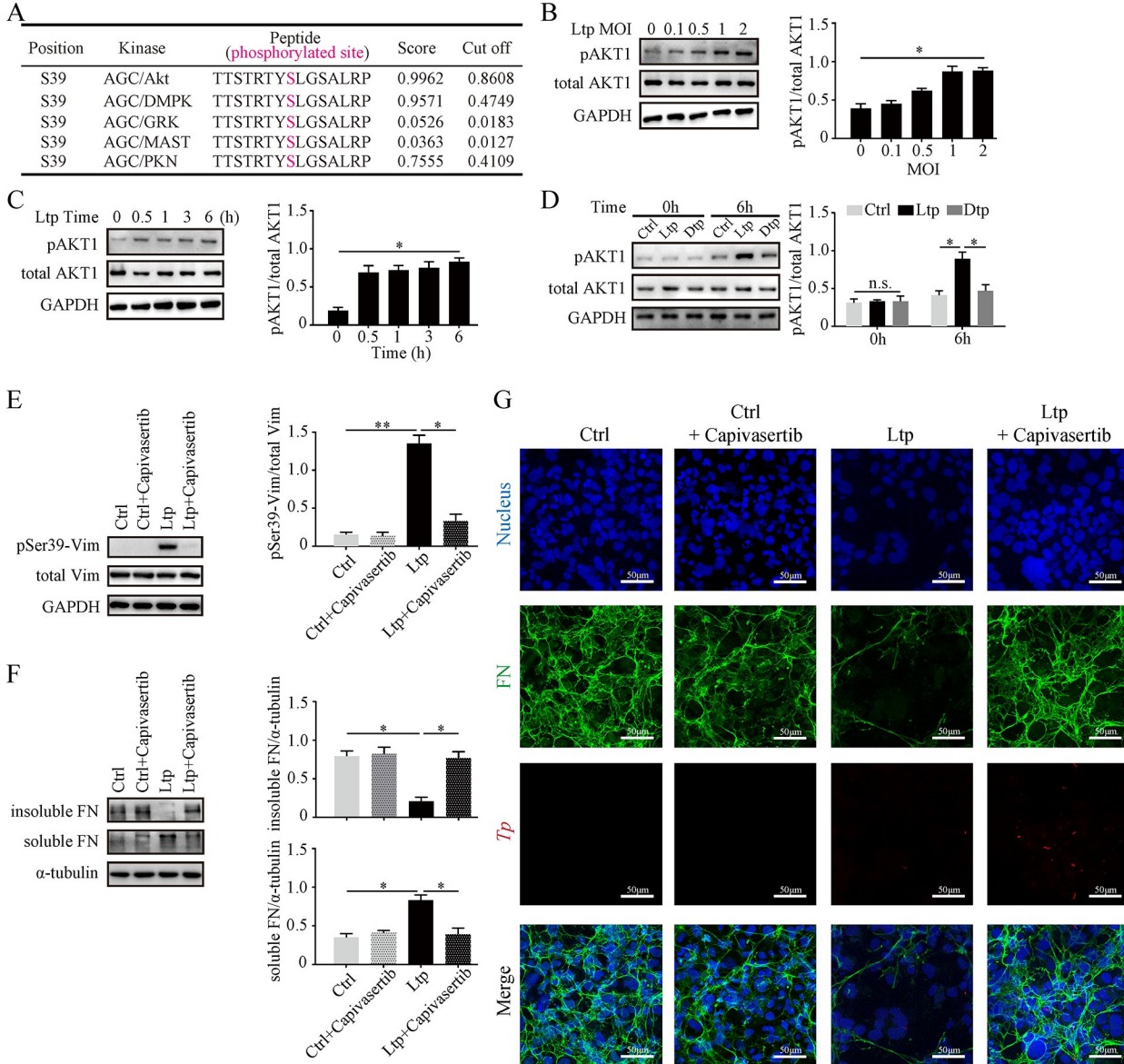

**Fig 4. *Tp* infection resulted in Ser39 residue phosphorylation of vimentin via the pAKT1 pathway.** (A) Screening the potential candidates of the protein kinase AGC group for Ser39 phosphorylation of vimentin; as predicted by high score and cutoff value. (B, C) Protein expressions of phosphorylated AKT1 (pAKT1) and total AKT1 in HMEC-1 (B) after 6 hours of stimulation with live *Tp* at various MOIs (0, 0.1, 0.5, 1, and 2) and (C) after stimulation with live *Tp* (MOI 2) for different durations (0, 0.5, 1, 3, and 6 hours). (D) Protein expressions of pAKT1 and total AKT1 in HMEC-1 after 6 hours of stimulation with Ctrl, live *Tp* (MOI 2), and dead *Tp* (MOI 2). (E-G)Addition of the AKT1 phosphorylation inhibitor capivasertib (1.0 μM) 1 hour prior to live *Tp* (MOI 2) stimulation in HMEC-1, followed by protein expression examination of (E) Ser39-phosphorylated vimentin (pSer39-Vim) and total vimentin, (F) insoluble and soluble FN, and (G) immunofluorescence observation of FN matrix with blue for the nucleus, green for the FN matrix, and red for *Tp*, scale bar = 50 μm. Ctrl: negative control; Ltp: live *Tp*; Dtp: dead *Tp*; Vim: vimentin; FN: fibronectin; n.s.: no significance; *: p value < 0.05; **: p value < 0.01.

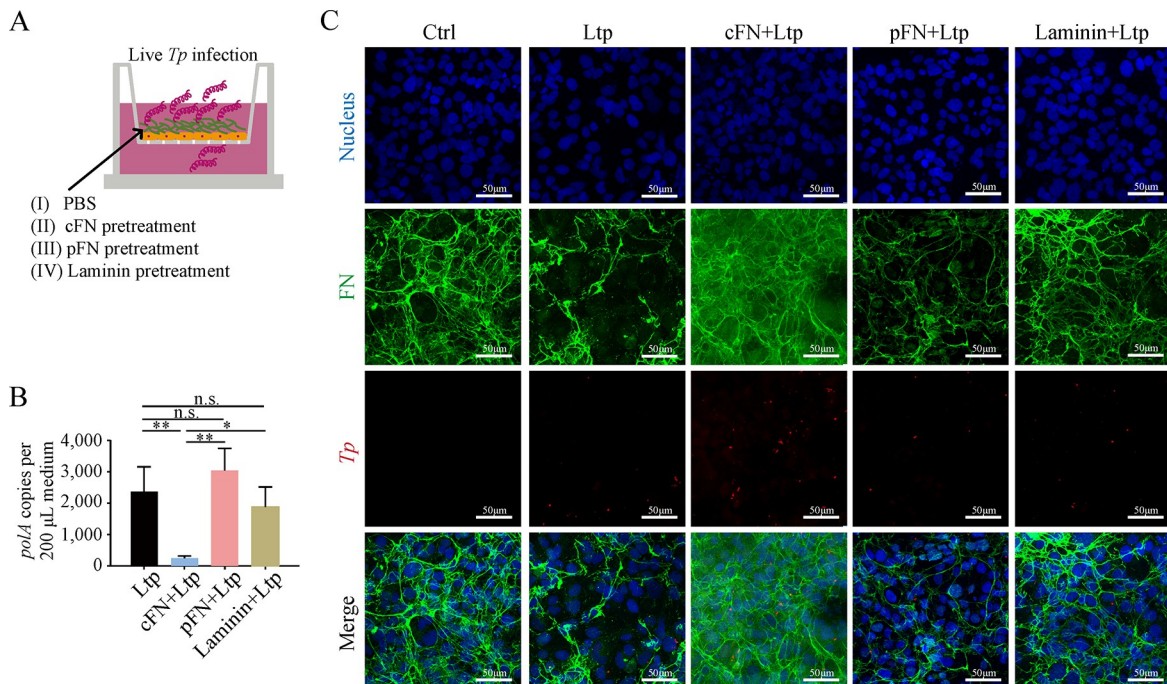

**Fig 5. cFN attenuated *Tp* penetration across the MEB. (A)** Schematic diagram of the endothelial barrier traversal experiment with additional ECM proteins; after the formation of HMEC-1 monolayer, 2.5 µg of each ECM protein added for 1 hour, and subsequent live *Tp* (MOI 2) added for stimulation. **(B)** Penetrated spirochete burden (copies of *Tp polA*) of all culture medium (200.0 µL) in the lower chamber; detected by qPCR. **(C)** Fibronectin matrix of HMEC-1 after as the same procedure to (A) in chamber slides; as observed by fluorescence microscopy; blue for the nucleus, green for the FN matrix, and red for *Tp*, scale bar = 50 µm. cFN: cellular fibronectin; pFN: plasma fibronectin; Ltp: live *Tp*; FN: fibronectin; n.s.: no significance; *: p value < 0.05; **: p value < 0.01.

proteins was predicted (**S1 File**), the function of additional cFN was then determined using the *Tp*-challenged rabbit model, as presented in **Fig 6A**. Initially, changes in *Tp* burden at the injection site were identified prior to the appearance of lesions since the inflammatory response would not disturb this phase. The results indicated that the cFN+Ltp group exhibited the highest in situ burden on Day 1 and Day 4 post-challenge, with a significant increase on Day 7 (**Fig 6B**), suggesting the possibility of a synergistic effect within the long-term interception and proliferation of *Tp*. Independently, there was no discernible impact of a single dose of each protein (cFN, pFN, or laminin, respectively) on the skin appearance of rabbits (**S5A Fig**), revealing no specific immune response of rabbit to heterologous proteins. However, the rabbits in each group began to develop rashes on Day 15 post-challenge (**S5B Fig**). The average area of cutaneous lesions in the cFN+Ltp group was constantly at the highest level (**Fig 6C**). Furthermore, when serological positivity for treponemal antibodies prevailed among the other groups (Day 11), the serum of the cFN+Ltp group remained antibody-negative, lagging in progress (**Table 1**). In contrast, the trend of non-treponemal antibodies in the cFN+Ltp group was more likely to turn positive and develop high titers (**Table 2**). Moreover, despite the darker coloration, increased fluctuation in sensation, and pronounced ulceration trend of skin lesions in the cFN+Ltp group on Day 26, there was no statistically significant difference in the in cutaneous *Tp* burden compared with the other groups (**Fig 6D**). Notably, pretreatment with pFN or laminin did not significantly affect *Tp* infection in the lymph node, liver, and spleen. However, cFN pre-treatment appeared to decrease organic infection, particularly in the lymph node (p value < 0.05) (**Fig 6E–6F**). When extrapolating the *Tp* burden from each biopsy to the entire organ by weight, the overall *Tp* load of liver or spleen and it would be much higher,

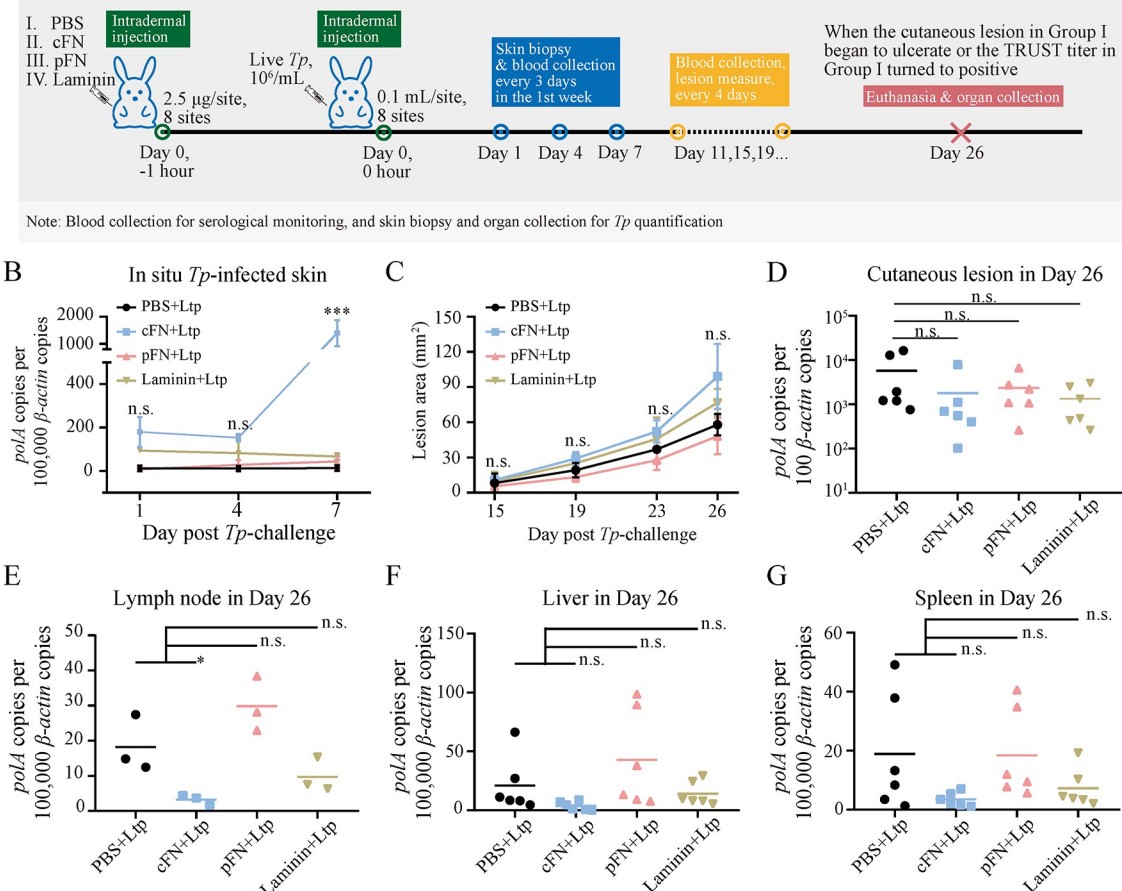

**Fig 6. cFN inhibited the cutaneous in situ dissemination of *Tp* and its organic infection *in vivo*. (A)** Timeline of the *in vivo Tp*-challenged experiment. **(B)** Spirochete burden (copies of *Tp polA* normalized to rabbit *β-actin*) of in situ *Tp*-infected skin (20 mg per sample) in the early period. **(C)** Average lesion area (mm²) post *Tp*-challenge. **(D-G)** Spirochete burdens (copies of *Tp polA* normalized to rabbit *β-actin*) of (D) cutaneous lesions (20 mg per sample, 2 biopsies per rabbit), (E) inguinal lymph nodes (one entire lymph node per rabbit, 25.0 mg per node), (F) livers (20 mg per sample, 2 biopsies per rabbit), and (G) spleens (10 mg per sample, 2 biopsies per rabbit) at the time of sacrifice (Day 26). cFN: cellular fibronectin; pFN: plasma fibronectin; n.s.: no significance; *: p value < 0.05; ***: p value < 0.001.

highlighting a decreasing trend in the cFN+Ltp group compared to the others. In conclusion, cFN stability can effectively limit the cutaneous in situ dissemination and organic infection of *Tp* to some extent.

## Discussion

FN has been recognized as a critical facilitator of pathogenic microorganism invasion. However, the specific mechanisms underlying *Tp* infection progression via FN attachment remain largely unexplored. Studies on the interactions between FN and other pathogenic microorganisms have indicated that ECM adhesion primarily triggers downstream colonization- and internalization-related mechanisms [36,37]. For instance, colonization followed by FN and laminin adhesion aided *Streptococcus uberis* in biofilm formation that developed antibiotic resistance [38]. *Staphylococcus aureus* was capable of activating an internalization response mediated by the interaction of its FN-binding adhesins with the integrin of the host cell. This mechanism assisted the pathogen in entering non-professional phagocytes to evade host immune clearance and hijack nutrients [39].

**Table 1. TPPA titer monitoring of treponemal antibodies in *Tp*-challenged rabbits.**

| Groups | | TPPA titer post-challenge | | | | | |
|---|---|---|---|---|---|---|---|
| | | D1/D4/D7 | D11 | D15 | D19 | D23 | D26 |
| PBS+Ltp | NO.1 | Neg. | 1:80 | 1:160 | 1:320 | 1:320 | 1:1280 |
| | NO.2 | Neg. | Neg. | 1:80 | 1:80 | 1:160 | 1:640 |
| | NO.3 | Neg. | 1:80 | 1:160 | 1:320 | 1:320 | 1:640 |
| cFN+Ltp | NO.1 | Neg. | Neg. | Neg. | 1:80 | 1:160 | 1:320 |
| | NO.2 | Neg. | Neg. | 1:80 | 1:320 | 1:320 | 1:320 |
| | NO.3 | Neg. | Neg. | 1:80 | 1:320 | 1:320 | 1:640 |
| pFn+Ltp | NO.1 | Neg. | 1:80 | 1:160 | 1:320 | 1:640 | 1:1280 |
| | NO.2 | Neg. | 1:80 | 1:160 | 1:160 | 1:320 | 1:1280 |
| | NO.3 | Neg. | 1:160 | 1:160 | 1:160 | 1:320 | 1:1280 |
| Laminin+Ltp | NO.1 | Neg. | 1:80 | 1:160 | 1:320 | 1:640 | 1:1280 |
| | NO.2 | Neg. | Neg. | 1:160 | 1:320 | 1:320 | 1:640 |
| | NO.3 | Neg. | 1:80 | 1:160 | 1:160 | 1:320 | 1:320 |

Neg.: negative; cFN: cellular fibronectin; pFN: plasma fibronection; Ltp: live *Tp*.

On the other hand, when infected with an equal dose of *Staphylococcus aureus*, osteoblasts with higher expression levels of FN and integrin exhibited reduced bacterial uptake than epithelial cells, while phagocytosis was promoted by FN knockdown [40]. This phenomenon did not contradict the above-mentioned enhanced bacterial internalization, which was explained by the inhibition of internalization by the matrix of FN fibrils surrounding osteoblasts. A similar scenario emerged in our study, in which *Tp* disrupted the ambient FN matrix by stimulating pSer39-vimentin within vascular endothelial cells. In both the *in vitro* endothelial barrier traversal experiment and *in vivo* dissemination experiment, the outcomes of the cFN+Ltp group (compared with those of the pFN+Ltp group or the Laminin+Ltp group) revealed a rescue of the FN matrix disaggregation due to cFN pretreatment, indicating that the dense FN matrix significantly hindered *Tp* from crossing the endothelial barrier and disseminating (**Figs 5 and 6**). In contrast, the addition of laminin or pFN failed to change the supramolecular

**Table 2. TRUST titer monitoring of treponemal antibodies in *Tp*-challenged rabbits.**

| Groups | | TRUST titer post-challenge | | | | | |
|---|---|---|---|---|---|---|---|
| | | D1/D4/D7 | D11 | D15 | D19 | D23 | D26 |
| PBS+Ltp | NO.1 | Neg. | Neg. | Neg. | Neg. | Neg. | Neg. |
| | NO.2 | Neg. | Neg. | Neg. | Neg. | Neg. | 1:1 |
| | NO.3 | Neg. | Neg. | Neg. | Neg. | Neg. | 1:2 |
| cFN+Ltp | NO.1 | Neg. | Neg. | Neg. | Neg. | Neg. | 1:2 |
| | NO.2 | Neg. | Neg. | Neg. | Neg. | USP | 1:8 |
| | NO.3 | Neg. | Neg. | Neg. | Neg. | Neg. | 1:4 |
| pFn+Ltp | NO.1 | Neg. | Neg. | Neg. | Neg. | Neg. | 1:1 |
| | NO.2 | Neg. | Neg. | Neg. | Neg. | Neg. | USP |
| | NO.3 | Neg. | Neg. | Neg. | Neg. | Neg. | Neg. |
| Laminin+Ltp | NO.1 | Neg. | Neg. | Neg. | Neg. | 1:2 | 1:8 |
| | NO.2 | Neg. | Neg. | Neg. | Neg. | Neg. | 1:8 |
| | NO.3 | Neg. | Neg. | Neg. | Neg. | Neg. | 1:2 |

Neg.: negative; USP: undiluted serum positive; cFN: cellular fibronectin; pFN: plasma fibronection; Ltp: live *Tp*.

structure of cFN, even though previous studies [15–17] have documented *Tp*'s adhesive effects on cFN, pFN, and laminin; however, our findings demonstrate that *Tp* exhibits maximal adherence to laminin instead of cFN (**S4 Fig**). Therefore, we established that while *Tp* can adhere to both laminin and FN, there exists a notable disparity in its ability to traverse the endothelial barrier. Simultaneously, cFN retained more *Tp* at the inoculated niches *in vivo*, exacerbating the severity of localized skin lesions and delaying the progression of systemic infection. These findings highlight the multifaceted role of FN matrix during the initial and deteriorated stages of infection and emphasize the complex interplay between pathogen invasion and host defense, at least for *Staphylococcus aureus* and *Tp*.

Thus, it's essential to distinguish the *Tp* dissemination mediated by FN matrix disaggregation and by FN adhesion. We acknowledge that *Tp* may utilize its surface proteins to bind FN and other ECM proteins, thereby promoting vascular interactions [41,42]. Due to the challenges associated with gene-editing for *Tp*, these investigations typically employ heterologous expression of *Tp* surface proteins in *Borrelia burgdorferi* to explore the functions of *Tp* proteins. However, this kind of intervention may not fully reflect the natural infection dynamics of *Tp* in our experiments: i) Whether these proteins exhibit the same adhesion properties under conditions of heterologous expression and natural infection remains a topic for further discussion, and no recent studies have demonstrated that overexpression or inhibition of *Tp* proteins can induce structural alterations in the FN matrix surrounding host cells. ii) The adhesion of *Tp* to specific ECM proteins results from the collective action of numerous surface proteins. We believe that FN matrix disaggregation-mediated and FN adhesion-mediated *Tp* dissemination are actually two independent pathways in accelerating the infection process.

When we keep focusing on FN matrix disaggregation, we recognized the involvement and significance of pAKT1/pSer39-vimentin pathway activated in *Tp* infection (**Fig 7**). It should be noted that the FN matrix disaggregation refers to changes in the matrix structure from dense

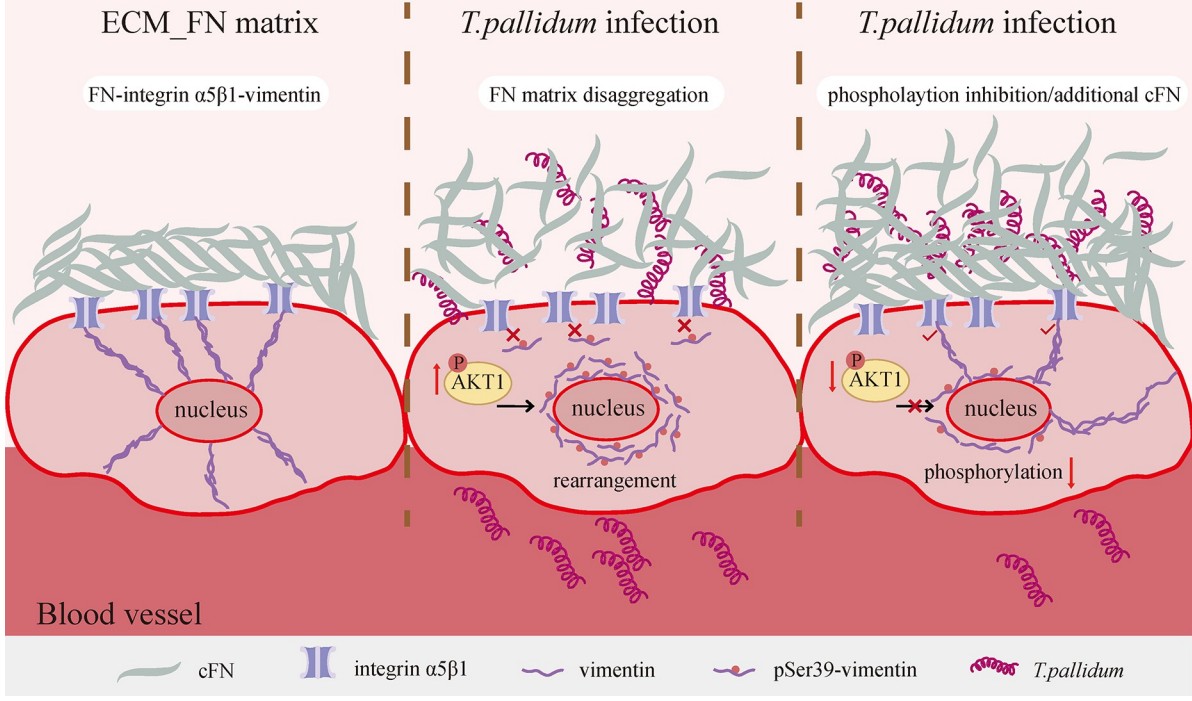

**Fig 7. Graphic summary of molecular mechanism.**

to sparse without altering the total FN content of monomers or dimers assembled into the matrix (such as **Fig 1A–1D**), while the lack of FN would lead to the almost disappearance of the matrix (**S6 Fig**). Our findings indicate that *Tp* activates this pathway by regulating post-translational modifications without affecting the overall protein levels of key molecules (total vimentin, total FN, and total AKT1 showed no significant changes after *Tp* stimulation), but inhibited expression of vimentin protein in HMEC-1 cells (HMEC-1$^{siVim}$) slightly reduced the FN protein level (**S7A Fig**), indicating that vimentin knockdown is unlikely to affect FN synthesis. However, the total FN significantly decreased in HMEC-1$^{siVim}$ after *Tp* infection (**S7B–S7C Fig**), which was different from the results in **Fig 1B–1D**. We speculated that *Tp* did not induce FN protein degradation or synthesis when vimentin expression was sufficient. However, knocking down vimentin reduced the total levels of various forms of vimentin, including the vimentin reserve that binds to integrins on the cell membrane to stabilize extracellular FN, which may disrupt the FN matrix by enhancing the matrix transformation pathway, likely leading to increased FN degradation. Furthermore, we showed that total vimentin levels remained unchanged while pSer39-vimentin decreased during *Tp* infection (**Fig 4E**), indicating that AKT1 impacts vimentin phosphorylation but not its synthesis or degradation. Knocking down AKT1 in HMEC-1 cells reduced total FN levels even without *Tp* stimulation, but did not affect total vimentin levels (**S8 Fig**), suggesting that AKT1 influences the protein level of total FN independently of vimentin phosphorylation, consistent with previous studies [43,44]. This suggests that knockdown of AKT1 is not entirely appropriate for our study, as it would result in a significant reduction in the most direct subject of study, the FN matrix, whether or not it was stimulated by *Tp*, leading to more confounders.

The results of our study also prompts a sudden and catastrophic quantitative change during *Tp* infection that: constrained by the FN matrix outside the cutaneous MEB, most *Tp* are subject to the rapid response of immune clearance pressure. In contrast, a minimal amount of *Tp* (MOI 2) is adequate to initiate a noticeable imbalance in vimentin phosphorylation and matrix disruption within a short period (1–3 hours). Meanwhile, pronounced deconstruction of the cFN matrix ensues promptly (within 6 hours), offering significant convenience for the rapid MEB traversal.

Spirochetes breached the in situ MEB in *Tp*-challenged rabbits within 24 hours after subcutaneous injection; however, the cFN+Ltp group had a more elevated burden than the other groups. Moreover, the cFN+Ltp group exhibited a relatively larger area of localized lesions, and the appearance of treponemal antibodies in peripheral blood occurred four days later than in the other groups. Considering our published data and previous literature suggesting that the non-treponemal (TRUST) antibodies are mediated by damaged host cell-released cardiolipin [45,46], rabbits were euthanized when TRUST began to turn positive in the PBS+Ltp group (Day 26) to minimize the effect of host cells. It is noteworthy that the cFN+Ltp group showed the highest TRUST titer on Day 26, accompanied by the lowest organic *Tp* burden and the most severe lesion ulcers, which may be attributed to the retention of excessive *Tp* at the injection niches. Some rabbits in the Laminin+Ltp group showed elevated TRUST titers; however, the lesion area and organic *Tp* burden were more similar to those of the PBS+Ltp group, likely due to the laminin adhesion with a portion of spirochetes. These findings confirm the notion that short-term maintenance of the stability of the FN matrix can effectively halt syphilitic progression.

Due to the variety of *Tp* proteins that facilitate adhesion to ECM and the dualistic role of cFN, we are likely to preserve a diminished phosphorylation level of vimentin within vascular endothelial cells during *Tp* infection. This strategy not only benefits endothelial barrier stabilization but also amplifies the *Tp* concentration sequestered within the infected niches, thereby significantly assisting in the immune eradication of this fragile microorganism.

## Supporting information

**S1 Fig. The viability and mortality of *Tp*. (A-B)** Motile *Tp* after stimulating HMEC-1 cells for (A) 6 hours and (B) 8 hours; observed by DFM under a 400-fold magnification, scale bar = 100 μm. **(C)** Rabbit back skin, 4 weeks after intradermal injection of live *Tp* isolated from the 6-hour co-culture medium. **(D)** Rabbit back skin, 4 weeks after intradermal injection of dead *Tp* inactivated by the co-cultured medium containing 1% (v/v) penicillin-streptomycin.
(TIF)

**S2 Fig. Impact of *Tp* on phosphorylation of serine residues at other sites of vimentin. (A-C)** Protein expressions of Ser56-phosphorylated (pSer56-) and total vimentin in HMEC-1 (A) after 6 hours of stimulation with live *Tp* at various MOIs (0, 0.1, 0.5, 1, and 2), (B) after stimulation with live *Tp* (MOI 2) for different durations (0, 0.5, 1, 3, and 6 hours), and (C) after 0 and 6 hours of stimulation with Ctrl, live *Tp* (MOI 2), and dead *Tp* (MOI 2). **(D-F)** Protein expressions of Ser72-phosphorylated (pSer72-) and total vimentin in HMEC-1 (D) after 6 hours of stimulation with live *Tp* at various MOIs (0, 0.1, 0.5, 1, and 2), (E) after stimulation with live *Tp* (MOI 2) for different durations (0, 0.5, 1, 3, and 6 hours), and (F) after 0 and 6 hours of stimulation with Ctrl, live *Tp* (MOI 2), and dead *Tp* (MOI 2). **(G-I)** Protein expressions of Ser83-phosphorylated (pSer83-) and total vimentin in HMEC-1 (G) after 6 hours of stimulation with live *Tp* at various MOIs (0, 0.1, 0.5, 1, and 2), (H) after stimulation with live *Tp* (MOI 2) for different durations (0, 0.5, 1, 3, and 6 hours), and (I) after 0 and 6 hours of stimulation with Ctrl, live *Tp* (MOI 2), and dead *Tp* (MOI 2). Ctrl: negative control; Ltp: live *Tp*; Dtp: dead *Tp*; Vim: vimentin.
(TIF)

**S3 Fig. Co-immunoprecipitation of vimentin and AKT1.** Cell lysates from negative control and live *Tp*-infected HMEC-1 cells after a 6-hour infection period were subjected to immunoprecipitation using an **(A)** anti-vimentin antibody or **(B)** an anti-AKT1 antibody, followed by Western blotting to detect the immunoprecipitated complexes of AKT1 and vimentin. The IgG antibody was used to demonstrate the absence of non-specific binding, and the input lysates (10% of the total lysate) were probed to confirm the presence of both proteins in the lysates. Ctrl: negative control; Ltp: live *Tp*; IP: immunoprecipitation; IB: immunoblotting; Vim: vimentin.
(TIF)

**S4 Fig. The adhesion capability of *Tp* to different ECM proteins. (A)** Nonadherent *Tp* (red arrow) in the supernatant combined with those washed down by PBS, after *Tp*'s adhesion to cFN, pFN, or laminin for 6 hours; observed by DFM under a 400-fold magnification, scale bar = 100 μm. **(B)** Quantification of non-adherent *Tp* per field of view under a 400-fold magnification using DFM. cFN: cellular fibronectin; pFN: plasma fibronectin; n.s.: no significance; ***: p value < 0.001.
(TIF)

**S5 Fig. Changes in cutaneous lesions in *Tp*-challenged rabbits. (A)** The appearance of rabbit's skin after the single use of different ECM proteins or PBS, respectively. **(B)** Changes in *Tp*-challenged rabbit's skin at various time points from no evident signs (Day 11) to obvious redness and ulcer (Day 15, 19, and 23) and to sacrifice (Day 26). cFN: cellular fibronectin; pFN: plasma fibronectin; Ltp: live *Tp*.
(TIF)

**S6 Fig. Western blotting and immunofluorescence analysis of HMEC-1 cells with fibronectin knockdown by siRNA. (A)** Protein experssions of total fibronectin in HMEC-1 cells with

fibronectin knockdown by siRNA (HMEC-1$^{siFN}$) and cells with scramble siRNA interference as negative control (HMEC-1$^{siNC}$), respectively. **(B)** Protein experssions of total fibronectin in HMEC-1 $^{siFN}$ and HMEC-1$^{siNC}$ after stimulation with live *Tp* (MOI 2) for 6 hours, respectively. **(C)** Fibronectin matrix of HMEC-1$^{siFN}$ and HMEC-1$^{siNC}$ after stimulation with live *Tp* (MOI 2) for 6 hours, respectively; observed by fluorescence microscopy, blue for the nucleus, green for the FN matrix, red for *Tp*, scale bar = 100 μm. NC: negative control; FN: fibronectin; Ltp: live *Tp*.
(TIF)

**S7 Fig. Western blotting and immunofluorescence analysis of HMEC-1 cells with vimentin knockdown by siRNA. (A)** Protein experssions of pSer39-vimentin, total vimentin, and total fibronectin in HMEC-1 cells with vimentin knockdown by siRNA (HMEC-1$^{siVim}$) and cells with scramble siRNA interference as negative control (HMEC-1$^{siNC}$), respectively. **(B)** Protein experssions of pSer39-vimentin, total vimentin, and total fibronectin in HMEC-1$^{siVim}$ and HMEC-1$^{siNC}$ after stimulation with live *Tp* (MOI 2) for 6 hours, respectively. **(C)** Fibronectin matrix of HMEC-1$^{siVim}$ and HMEC-1$^{siNC}$ after stimulation with live *Tp* (MOI 2) for 6 hours, respectively; as observed by fluorescence microscopy; blue for the nucleus, green for the FN matrix, and red for *Tp*, scale bar = 100 μm. NC: negative control; Vim: vimentin; FN: fibronectin; Ltp: live *Tp*.
(TIF)

**S8 Fig. Western blotting and immunofluorescence analysis of HMEC-1 cells with AKT1 knockdown by siRNA. (A)** Protein experssions of AKT1, vimentin, and fibronectin in HMEC-1 cells with AKT1 knockdown by siRNA (HMEC-1$^{siAKT1}$) and cells with scramble siRNA interference as negative control (HMEC-1$^{siNC}$), respectively. **(B)** Protein experssions of pSer39-vimentin, total vimentin, and fibronectin in HMEC-1 $^{siAKT1}$ and HMEC-1$^{siNC}$ after stimulation with live *Tp* (MOI 2) for 6 hours, respectively. **(C)** Fibronectin matrix of HMEC-1 $^{siAKT1}$ and HMEC-1$^{siNC}$ after stimulation with live *Tp* (MOI 2) for 6 hours, respectively; observed by fluorescence microscopy, blue for the nucleus, green for the FN matrix, red for *Tp*, scale bar = 100 μm. NC: negative control; Vim: vimentin; FN: fibronectin; Ltp: live *Tp*.
(TIF)

**S1 Table. Screening the potential serine/threonine phosphorylation sites on vimentin and their associated protein kinases.**
(DOCX)

**S1 Movie. Video of *Tp* mobility after 6-hour co-culture with HMEC-1 cells, corresponding to S1A Fig.**
(MP4)

**S2 Movie. Video of *Tp* mobility after 8-hour co-culture with HMEC-1 cells, corresponding to S1B Fig.**
(MP4)

**S1 File. Alignment and 3D structure prediction of FN_human and FN_rabbit. (A)** Alignment by the whole sequences of FN_human and FN_rabbit. **(B)** Alignment by the individual domains of FN_human and FN_rabbit and the 3D structure prediction.
(PDF)

**S2 File. Methods for gene knockdown and the sequences of small interfering RNAs (siRNAs).**
(PDF)

**S1 Data. Excel spreadsheet containing, in separate sheets, the underlying numerical data and statistical analysis for Figs 1B, 1C, 1D, 1E, 1F, 1G, 2A, 2B, 2C, 2E, 2F, 2G, 3B, 3E, 4B, 4C, 4D, 4E, 4F, 5B, 6B, 6C, 6D, 6E, 6F, 6G, and S4B.**
(XLSX)

## Acknowledgments

We thank Dr. Wentao Chen (Dermatology Hospital, Southern Medical University) for providing the *Tp polA* qPCR standard and Ms. Jialin Huang (Dermatology Hospital, Southern Medical University) for providing the Rabbit *β-actin* qPCR standard. And we appreciate Dr. Jun Liu (Dermatology Hospital, Southern Medical University) for manuscript grammar editing.

## Author Contributions

**Conceptualization:** Xi Luo, Litian Zhang, Yinbo Jiang, Wujian Ke.

**Data curation:** Xi Luo, Litian Zhang, Xiaoyuan Xie.

**Investigation:** Xi Luo, Litian Zhang, Xiaoyuan Xie, Liyan Yuan, Yanqiang Shi.

**Project administration:** Xi Luo, Litian Zhang.

**Resources:** Xi Luo, Litian Zhang, Xiaoyuan Xie, Yinbo Jiang.

**Supervision:** Yinbo Jiang, Wujian Ke, Bin Yang.

**Writing – original draft:** Xi Luo, Litian Zhang.

**Writing – review & editing:** Xi Luo, Litian Zhang, Xiaoyuan Xie, Liyan Yuan, Yanqiang Shi, Yinbo Jiang, Wujian Ke, Bin Yang.

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
