## [Decision Letter · Decision Letter 0]

1 May 2024

Dear Prof. Yang,

Thank you very much for submitting your manuscript "Phosphorylated vimentin-triggered fibronectin matrix disaggregation enhances the dissemination of Treponema pallidum subsp. pallidum across the microvascular endothelial barrier" for consideration at PLOS Pathogens. As with all papers reviewed by the journal, your manuscript was reviewed by members of the editorial board and by several independent reviewers. In light of the reviews (below this email), we would like to invite the resubmission of a significantly-revised version that takes into account the reviewers' comments.

Thank you for submitting your manuscript to PLOS Pathogens. Your study is intriguing and contributes novel insights to the field. After careful consideration, I recommend that the manuscript undergo major revisions.

In addressing the concerns raised, it is crucial to clearly differentiate between the mechanisms of cFN blockade and cFN adhesion affecting Tp's trans-endothelial migration. Your response should include detailed mechanistic explanations or experimental data to support the hypothesis that Tp penetration is inhibited by cFN blockade rather than adhesive interactions. Please consider conducting additional experiments or providing theoretical rationale backed by literature.

Another critical issue is the absence of co-immunoprecipitation experiments for AKT and vimentin. The co-localization and potential interaction between these proteins may provide significant insights into the signaling pathways involved in your study's context. I recommend including these experiments to strengthen the manuscript and provide a more comprehensive understanding of the molecular mechanisms at play.

Please address these points thoroughly in your revised manuscript and provide a detailed point-by-point response to each of the reviewer’s comments. We look forward to receiving your revised submission.

We cannot make any decision about publication until we have seen the revised manuscript and your response to the reviewers' comments. Your revised manuscript is also likely to be sent to reviewers for further evaluation.

Sincerely,

yimou wu

Guest Editor

PLOS Pathogens

Matthew Wolfgang

Section Editor

PLOS Pathogens

Michael Malim

Editor-in-Chief

PLOS Pathogens

orcid.org/0000-0002-7699-2064

Thank you for submitting your manuscript to PLOS Pathogens. Your study is intriguing and contributes novel insights to the field. After careful consideration, I recommend that the manuscript undergo major revisions.

In addressing the concerns raised, it is crucial to clearly differentiate between the mechanisms of cFN blockade and cFN adhesion affecting Tp's trans-endothelial migration. Your response should include detailed mechanistic explanations or experimental data to support the hypothesis that Tp penetration is inhibited by cFN blockade rather than adhesive interactions. Please consider conducting additional experiments or providing theoretical rationale backed by literature.

Another critical issue is the absence of co-immunoprecipitation experiments for AKT and vimentin. The co-localization and potential interaction between these proteins may provide significant insights into the signaling pathways involved in your study's context. I recommend including these experiments to strengthen the manuscript and provide a more comprehensive understanding of the molecular mechanisms at play.

Please address these points thoroughly in your revised manuscript and provide a detailed point-by-point response to each of the reviewer’s comments. We look forward to receiving your revised submission.

Reviewer's Responses to Questions

**Part I - Summary**

Reviewer #1: In the present study, the authors reported that vimentin may disrupt cFN-integrin binding after Tp infection, facilitating the dissemination of Tp across the MEB. phosphorylated Ser39 of vimentin triggered fibronectin matrix disaggregation and the related mechanisms and domestrated that the stability of the cFN matrix alleviated the dissemination of Tp across the MEB. However, the manuscript need more revision before publication.

Reviewer #2: The authors present a compelling case that one effect of Treponema pallidum (Tp) infection involves AKT-1 activation and its effects on vimentin filament organization leading to disruption of vascular extracellular matrix (ECM). Vimentin is known to be a stress protein (e.g. see Pattabiraman et al, 2020. Vimentin protects differentiating stem cells from stress. Scientific Reports, 10(1), p.19525) so a range of changes in cellular inputs are likely to be influenced by the state of the vimentin filament network.

The observations seem compelling and medically relevant.

Reviewer #3: In this manuscript, the authors demonstrate through in vivo and in vitro experiments that live Tp activates pAKT1/pSer39 vimentin signal to expedite the disaggregation of the FN matrix and provide a new perspective on the mechanism by which Tp disrupts the ECM. This manuscript is logically clear, with fluent and understandable language and a detailed methodological presentation. Some details need attention, and it is ready for acceptance after minor modifications.

Reviewer #4: In this study, Luo et al. described a novel molecular basis of T. pallidum to penetrate endothelial cells and aimed to correlate this mechanism to T. pallidum dissemination. While the concept is interesting, some experimental design allows only get the correlation of phenotypes rather than building a cause and results relationship of these phenotypes. Specifically, the current experimental design in the in vivo work would introduce many confounding factors, leading to the results unsupportive to the conclusion made by the authors. Additionally, the writing and clarity may need to be significantly improved by providing more details of how the experiments are performed. My suggestions are as below:

**Part II – Major Issues: Key Experiments Required for Acceptance**

Reviewer #1: Major Comments:

1. In this study, to investigate the effect of Tp on microvascular endothelial cells, HMEC-1 cells were co-culture with Tp in vitro to simulate the infection environment of syphilis. However, to our knowledge, the technology of Tp in vitro culture is not yet mature. How can you ensure that Tp can play a role similar to in vivo infection during co-culture with HMEC-1 cells. In addition, although Tp can survive in vitro for a short time, whether it has corresponding infective activity remains unproven.

2. Different MOL of Tp were used for this study, but the specific quantitative method of Tp did not shown to be mentioned in the material method.

3. Result 4. There was no significant difference in PAKT1 level after Tp stimulation with MOL 1 and 2 in Fig4B. Why should the stimulation concentration with Mol 2 be used in subsequent experiments.

4. The Figure of Wb in Fig4F is not clear, please replace it with a clearer Figure.

5. Please detect the stability of cFN matrix and the spread of Tp infection after inhibition the phosphorylation of pSer39.

6. Result 6. Is direct injection of cFN via intradermal injection similar to vaccination? How to ensure the functional structure of the protein in vivo after injected directly without any adjuvants? In addition, The PCR results in different tissues showed that the load of Tp in the PBS+Ltp group was also not high, Whether this indicates that Tp does not diffuse effectively in vivo, which will directly affect the reliability of this result.

Reviewer #2: These studies were carried out in human HMEC-1 cells. A key experiment that seems missing is to determine how vimentin-null HMEC-1 cells respond to Tp infection. The authors could then determine how the virus-driven over-expression of wild type vimentin or S39A vimentin (in a vimentin null background) influences the patterns of gene expression and the Tp effect on fibronectin/ECM. In a similar way, it would be interesting to know how living and dead Tp influence patterns of gene expression.

Generating Vim-/- HMEC-1 cell lines should be a straightforward using CRISPR-Cas9; their immunofluorescence microscopy assay would enable them to quickly establish whether it is vimentin or the disruption of normal vimentin organization, in response to AKT-1 (e.g. by VimS39A expression) that is involved in the cell's response to Tp.

I am not commenting on the rabbit experiments, beyond my expertise.

Reviewer #3: 1. The authors employed both live and deceased Tp in the study, did there any evidence supporting the viability and mortality of Tp.

Reviewer #4: 1. Most of the conclusions were made based on the correlation of two different phenotypes. The examples are as follows:

(1) (line 283-310), The results showed the correlation of the levels of Vimentin with the levels of cFN. Such correlations do not necessarily suggest “TP-induced vimentin phosphorylation and rearrangement promoted disaggregation if cFN” concluded in the line 281-282. To directly tie vimentin into the contributor of the cFN pathways, a vimentin-knock down cells would need to be added in the same experimental setup of infection with Ltp.

(2) (line 311-328), These results showed the correlation of the levels of AKT1 and cFN. Such correlations do not necessarily suggest “Tp facilitated the phosphorylation of Ser39-vimentin via pAKT1 pathway” listed in line 311. AKT1 needs to be knocked down in the cells to directly tie AKT1 to cFN.

(3) (line 329-351) These results aimed to demonstrate the title in line 329 “Dense cFN matrix attenuated Tp penetration through the trans-endothelial barrier in vitro.” In order to make such conclusions, the experimental design should include the cells deficient of cFN (i.e., cFN-knockdown cells) and examine the Tp penetration. The current experimental design does not lead to such a conclusion.

These weaknesses of experimental design are major, leading to the current results to be unsupportive for the conclusions the authors would like to make.

2. In line 352 to 374, the authors attempted to pre-incubated cFN with T. pallidum and inoculate rabbits with cFN-T. pallidum mixtures and compared the resulting T. pallidum infectivity with the infectivity from the untreated T. pallidum. As there is always some unbound cFN in the mixtures of cFN and T. pallidum, the unbound cFN may have some physiological functions. Therefore, it is unclear whether the dissemination inhibition in cFN-treated T. pallidum is due to the binding of cFN to T. pallidum or the physiological functions caused by unbound cFN. pallidum. Therefore, such experimental design for the in vivo experiments introduced many confounding factors, which does not necessarily support the conclusion that “cFN inhibited the cutaneous in situ dissemination of Tp and its organic infection in vivo” in line 353.

3. The model that the authors would like to test in this study is to tie fibronectin disaggregation triggered by T. pallidum to T. pallidum transmigration through microvascular endothelial cells. However, the other model that was also raised to explain the transmigration and vascular adhesion is that T. pallidum by using its surface protein to bind to fibronectin to promote such vascular interactions (PMID28484210, 27683203). Therefore, the dissemination phenotypes observed in vivo and the transmigration observed in trans-well can be caused by this adhesion model rather than the model the authors proposed. Before additional experiments differentiate the fibronectin disaggregation-mediated vs. fibronectin adhesion-mediated T. pallidum dissemination/transmigration, it is difficult to make the conclusion that these results support the title of this manuscript.

**Part III – Minor Issues: Editorial and Data Presentation Modifications**

Reviewer #1: Minor Comments:

1. In the section of introduction, the description of research progresss in Fibronectin, extracellular matrix, and microvascular endothelial barrier is too simple and lacks logic, which makes it difficult to agree with the necessity of this study.

2. Line 213, ml should be written in the same way as before.

3. Lin133, and 200, please replace “in vivo” with “in vivo”.

4. The red fluorescence of Tp in Fig1A is not obvious.

5. Fig3C: Similarly, Less and not obvious Tp red fluorescence in this figure will affect the reliability of the experiment

Reviewer #2: (No Response)

Reviewer #3: 2. In line 204, I noticed that the decimal place in “100 μL” is not consistent with other numbers. It would be beneficial to ensure uniformity in reporting the decimal places for better clarity and accuracy.

3. In Figure 1-6, there is an asterisk (*) or two asterisk (*) symbol, but there is no specific explanation provided in the figure caption. It would be helpful to include a clear description of what the asterisk represents in the figure.

Reviewer #4: 1. Generally, the writing of figure legends and result section needs to be significantly improved. The figure legends need to include sufficient details of how the experiments are performed. The result section needs to start from describing briefly how the work is performed.

2. The introduction section requires to provide sufficient background the authors need to know to understand this work. The current version of introduction is not very clear. For example, it is better to discuss more details about the differences of cellular and soluble plasma FN as such differences drive the implication of the key in vivo experiment by introducing different FN to demonstrate T. pallidum penetrating mechanisms. Further, does T. pallidum bind to fibronectin? How much is known about that? In Line 92, where is that Ser39 coming from? The detailed introduction of Ser39 is needed.

3. (Line 35), Typo …. test “demonstrated” that

4. Throughout the manuscript, the quality of the immunofluorescent figures was very low. Additionally, no bacteria can be identified in the panel showing the staining results of T. pallidum.

5. (Line 259-260), Is there any way to quantify the clustered cFN matrix and try to do some statistical analysis?

6. (Fig. 1B to D) Are these FN showing actually the sum of cFN and pFN so the conclusion that cFN levels are not altered is not correct? This is very important for the author to make the conclusions of certain FN making the contribution of particular phenotypes.

7. (Line 276, soluble and insoluble FN), The verification of no pFN is present at detectable levels in cFN would need to be provided in order for the authors to make the conclusion in the following studies that they are actually studying cFN (rather than the mixture of pFN and cFN).

8. (Line 284), What is “its”? Is this the solubility of vimentin or FN? Additionally, in the same line, which protein is inhibited for its translocation? Vimentin or FN?

9. (Fig. 2D) Please provide the arrow to specify the location of "confined vimentin distribution around the nucleus." Additionally, there seems to be one cell in the field of the cells treated with LTp. please show the field with more cells as the represented picture.

10. (Line 317), What is AKT? This needs to be introduced in the introduction section. Is AKT AGC kinase?

11. (Line 354-374), in the in vivo experiment, the cFN, laminin inoculated into rabbits were not from rabbits. How similar are the sequences from human vs. rabbit cFN or human vs. laminin? Would it be possible that those proteins trigger anti-cFN, or anti-laminin antibodies that leads to the dissemination blocking results you observed?

PLOS authors have the option to publish the peer review history of their article (what does this mean?). If published, this will include your full peer review and any attached files.

Reviewer #1: No

Reviewer #2: No

Reviewer #3: No

Reviewer #4: No
---

## [Decision Letter · Decision Letter 1]

5 Aug 2024

Dear Prof. Yang,

We are pleased to inform you that your manuscript 'Phosphorylated vimentin-triggered fibronectin matrix disaggregation enhances the dissemination of *Treponema pallidum subsp. pallidum* across the microvascular endothelial barrier' has been provisionally accepted for publication in PLOS Pathogens.**

*Before your manuscript can be formally accepted you will need to complete some formatting changes, which you will receive in a follow up email. A member of our team will be in touch with a set of requests.*

*Please note that your manuscript will not be scheduled for publication until you have made the required changes, so a swift response is appreciated.*

*IMPORTANT: The editorial review process is now complete. PLOS will only permit corrections to spelling, formatting or significant scientific errors from this point onwards. Requests for major changes, or any which affect the scientific understanding of your work, will cause delays to the publication date of your manuscript.*

*Should you, your institution's press office or the journal office choose to press release your paper, you will automatically be opted out of early publication. We ask that you notify us now if you or your institution is planning to press release the article. All press must be co-ordinated with PLOS.*

*Thank you again for supporting Open Access publishing; we are looking forward to publishing your work in PLOS Pathogens.*

*Best regards,*

yimou wu

Guest Editor

PLOS Pathogens

Matthew Wolfgang

Section Editor

PLOS Pathogens

Michael Malim

Editor-in-Chief

PLOS Pathogens

orcid.org/0000-0002-7699-2064

Reviewer Comments (if any, and for reference):

*Reviewer's Responses to Questions*

*
**Part I - Summary**
*

*Please use this section to discuss strengths/weaknesses of study, novelty/significance, general execution and scholarship.*

*Reviewer #1: This investigation revealed the active pAKT1/pSer39-vimentin*

signal triggered by live *Tp* to expedite the disaggregation of the FN matrix and**

highlighted the importance of FN matrix stability in syphilis, thereby providing a novel

perspective on ECM disruption mechanisms that facilitate *Tp* dissemination**

*across the MEB.*

*Reviewer #2: In their response, the authors reported their attempts to generate Vimentin null cells - They should include a description of their observations in the revised manuscript.*

*Reviewer #3: The author has made corresponding answers to the comment, and the revised manuscript has been greatly improved.*

*
**Part II – Major Issues: Key Experiments Required for Acceptance**
*

*Generally, there should be no more than 3 such required experiments or major modifications for a "Major Revision" recommendation. If more than 3 experiments are necessary to validate the study conclusions, then you are encouraged to recommend "Reject".*

*Reviewer #1: The authors have addressed all of the review comments about the previous manuscript. I believe that the manuscript is ready for publication in this journal.*

*Reviewer #2: (No Response)*

*Reviewer #3: (No Response)*

*
**Part III – Minor Issues: Editorial and Data Presentation Modifications**
*

*Please use this section for editorial suggestions as well as relatively minor modifications of existing data that would enhance clarity.*

*Reviewer #1: (No Response)*

*Reviewer #2: (No Response)*

*Reviewer #3: (No Response)*

*PLOS authors have the option to publish the peer review history of their article (what does this mean?). If published, this will include your full peer review and any attached files.*

**

*Reviewer #1: No*

*Reviewer #2: No*

*Reviewer #3: No*

---

## [Editor Report · Acceptance letter]

21 Aug 2024

Dear Prof. Yang,

We are delighted to inform you that your manuscript, "Phosphorylated vimentin-triggered fibronectin matrix disaggregation enhances the dissemination of <i>Treponema pallidum subsp. pallidum<i> across the microvascular endothelial barrier," has been formally accepted for publication in PLOS Pathogens.

Best regards,

Michael Malim

Editor-in-Chief

PLOS Pathogens

orcid.org/0000-0002-7699-2064